# BMP7 increases protein synthesis in SW1353 cells and determines rRNA levels in a NKX3-2-dependent manner

Ellen G. J. Ripmeester[1], Tim J. M. Welting[1,2], Guus G. H. van den Akker[1], Don A. M. Surtel[1], Jessica S. J. Steijns[1], Andy Cremers[1], Lodewijk W. van Rhijn[1,2], Marjolein M. J. Caron[1]*

**1** Laboratory for Experimental Orthopedics, Department of Orthopedic Surgery, Maastricht University, Maastricht, the Netherlands, **2** Laboratory for Experimental Orthopedics, Department of Orthopedic Surgery, Maastricht University Medical Center, Maastricht, the Netherlands

* marjolein.caron@maastrichtuniversity.nl

**Data Availability Statement:** The minimal data set is available from the figshare database (DOI: 10.6084/m9.figshare.19071530).

## Abstract

BMP7 is a morphogen capable of counteracting the OA chondrocyte hypertrophic phenotype via NKX3-2. NKX3-2 represses expression of RUNX2, an important transcription factor for chondrocyte hypertrophy. Since RUNX2 has previously been described as an inhibitor for 47S pre-rRNA transcription, we hypothesized that BMP7 positively influences 47S pre-rRNA transcription through NKX3-2, resulting in increased protein translational capacity. Therefor SW1353 cells and human primary chondrocytes were exposed to BMP7 and rRNA (18S, 5.8S, 28S) expression was determined by RT-qPCR. NKX3-2 knockdown was achieved via transfection of a NKX3-2-specific siRNA duplex. Translational capacity was assessed by the SUNsET assay, and 47S pre-rRNA transcription was determined by transfection of a 47S gene promoter-reporter plasmid. BMP7 treatment increased protein translational capacity. This was associated by increased 18S and 5.8S rRNA and NKX3-2 mRNA expression, as well as increased 47S gene promotor activity. Knockdown of NKX3-2 led to increased expression of RUNX2, accompanied by decreased 47S gene promotor activity and rRNA expression, an effect BMP7 was unable to restore. Our data demonstrate that BMP7 positively influences protein translation capacity of SW1353 cells and chondrocytes. This is likely caused by an NKX3-2-dependent activation of 47S gene promotor activity. This finding connects morphogen-mediated changes in cellular differentiation to an aspect of ribosome biogenesis via key transcription factors central to determining the chondrocyte phenotype.

## Introduction

In healthy articular cartilage, the chondrocytes' catabolic and anabolic processes involving maintenance of the cartilage extracellular matrix (ECM) are balanced [1, 2]. However, during osteoarthritis (OA) progression, this homeostasis is disrupted and presents with a hypertrophic chondrocyte phenotype, resulting in the active degradation of the ECM [3, 4]. Previously

**Funding:** This work was financially supported by the Dutch Arthritis Association (https://reumanederland.nl; grants LLP14, 13-02-201, 15-03-403) and Stichting de Weijerhorst (www.deweijerhorst.nl) granted to TW and LvR. The funders had no role in study design, data collection, and analysis, decision to publish, or preparation of the manuscript.

**Competing interests:** MMJ Caron and TJM Welting are inventors on patents WO2017178251 and WO2017178253 (Owned by Chondropeptix). LW van Rhijn and TJM Welting are shareholder in Chondropeptix and are CDO, and CSO of Chondropeptix, respectively. All other authors have no competing interests to declare. This does not alter our adherence to PLOS ONE policies on sharing data and materials.

our group and others demonstrated that bone morphogenetic protein 7 (BMP7) is able to beneficially counteract this chondrocyte hypertrophic phenotype in OA [5–8]. This phenotypic change is characterized by increased chondrogenic gene and protein expression, while the expression of hypertrophic-, cartilage degrading and inflammatory factors is reduced in OA chondrocytes following BMP7 exposure [5–8].

The ribosome is the central player in the cell's protein translational machinery. The mammalian ribosome consists of two subunits (40S and 60S). Together these subunits contain approximately 80 different ribosomal proteins and 4 ribosomal RNAs (rRNA) [9]. The rRNAs are essential for the ribosome's translation capacity [10]. Transcription of these rRNAs is a major rate-limiting step in the biogenesis of ribosomes [11, 12]. Three out of four rRNAs are transcribed as a large multi-cistronic 47S precursor by the dedicated RNA Polymerase I [11]. The primary 47S transcript undergoes multiple endo- and exoribonucleolytic cleavage steps, as well as a high number of post-transcriptional modifications [13]. Eventually, the 47S pre-rRNA forms the mature 18S, 5.8S and 28S rRNAs [13].

Aberrant 47S ribosomal DNA (rDNA) gene transcription results in cell-phenotype changes and has been described in relation to normal cell differentiation as well as disease [14–18]. Indeed, ribosome biogenesis is tightly regulated and control of rDNA gene transcription has been described for epidermal growth factor (EGF) [11, 19], insulin-like growth factor-1 (IGF-1) [20–22], and serum [11]. The primary transcriptional targets of these growth factors differ, but all eventually result in modulation of rDNA gene transcription and thus influence ribosome biogenesis. Essential for the hypertrophy-suppressive action of BMP7 on OA chondrocytes is NK3 homeobox 2 (NKX3-2, also known as bagpipe homeobox homolog 1 (Bapx1)) [5, 23, 24]. NKX3.2 represses the expression of runt-related transcription factor 2 (RUNX2), an important transcription factor determinant for chondrocyte hypertrophy [4, 25]. Importantly, RUNX2 was previously demonstrated to inhibit 47S pre-rRNA transcription in osteocytic cells via interaction with HDAC1 (Histone deacetylase 1) and UBTF (Nucleolar transcription factor) [26, 27].

Although BMP7 can beneficially influence the chondrocyte phenotype with increased expression of cartilage ECM-proteins [4, 5, 23–25], its role in chondrocyte translational capacity and ribosome biogenesis has not been investigated before. Transcription of the 47S rRNA precursor is, at least in part, under control of RUNX2 [26–28]. Since BMP7 has been demonstrated to reduce RUNX2 expression levels in chondrocytes via NKX3-2 [5], we therefore hypothesized that BMP7 induces rRNA transcription in a NKX3-2-dependent manner. In this study we investigated the relation between BMP7 and 47S rRNA transcription and the involvement of NKX3.2.

## Materials and methods

### Cell culture

SW1353 cells [29] (ATCC HTB-94, STR profiled, Middlesex, UK) were cultured in a humidified atmosphere (37°C, 5% $CO_2$) in medium consisting of Dulbecco's minimal essential medium (DMEM)F12 (Life Technologies, Waltham, Massachusetts, USA) supplemented with 10% fetal calf serum (FCS; Sigma-Aldrich, Dorset, UK) and 1% penicillin/streptomycin (P/S, Invitrogen Life Technologies). Human articular chondrocytes (HACs) were isolated from cartilage obtained from total knee arthroplasty of end-stage (K&L grade 3–4) OA patients. Medical ethical permission was received from the Maastricht University Medical Center medical ethical committee; approval ID: MEC 2017–0183. Informed consent was obtained from all the participants and all methods were performed in accordance with the relevant guidelines and regulations. Chondrocytes were isolated with collagenase as previously described [23]. HACs

were cultured in DMEM/F12, complemented with 10% FCS, 1% Antibiotic/antimycotic (Invitrogen Life Technologies) and 1% NEAA (Life Technologies) under a humidified atmosphere (37˚C, 5% $CO_2$) until passage 2. Cells (30.000 cells/cm$^2$) were exposed to 1 nM BMP7 (R&D Systems, Minneapolis, Minnesota, USA) or 10 ng/ml Actinomycin D (Sigma-Aldrich) for 24 hours.

## Transfection of small interfering RNAs (siRNAs) and overexpression vectors

Transfection of SW1353s or human primary chondrocytes (30.000 cells/cm$^2$) with 100 nM of NKX3-2 siRNA duplex or a scrambled negative control siRNA (Control RNAi; small interfering RNA) was performed according to the manufacturer's protocol, using HiPerfect (Qiagen, Hilden, Germany). NKX3-2 and the Control siRNA duplexes were custom-made by Eurogentec (Liège, Belgium) and sequences are shown in Table 1. Treatment with BMP7 was started 5 hours post-transfection, and cells were harvested 24 hours post-stimulation for further analysis. The coding sequence of human NKX-3.2 was custom synthesized (GeneCust) with optimized codon usage and cloned into p3XFLAG-CMV-7.1 expression vector (Sigma-Aldrich). FLAG-NKX-3.2 was transiently overexpressed in SW1353 (by Fugene (Promega) according to the manufacturer's protocol) and human primary chondrocytes (by polyethyleneimine-mediated transfection) (1,000 ng of plasmid/well in 12-well plates). Cells were harvested after 24 hours.

## 47S rDNA promoter-reporter assay

The 47S rDNA gene promotor sequence was custom-made by Genecust, containing nucleotides -1000 up to +60 from the human 47S rDNA transcription start site of the 47S rDNA gene sequence (Ensembl), and cloned into the pNL1.2 vector (Promega, Madison, Wisconsin, USA). The pNL1.2_47S-rDNA promoter plasmid was transfected into SW1353 cells (30.000 cells/cm$^2$; n = 6 samples per condition) with Fugene6. Five hours Post-transfection, cells were incubated for 24 hours with 1 nM BMP7. Post-stimulation, cells were harvested for bioluminescence analysis using cell culture lysis buffer (Promega). Promotor-activity was measured with the Nano-Glo Luciferase Assay System (Promega) on a Tristar LB942 (Berthold Technologies, Bad Wildbad, Germany). Relative differences were determined as compared to control conditions following correction for background and normalization by DNA-content. DNA-content was measured using a SYBR-GREEN assay (Invitrogen).

## Protein concentration analysis

The protein concentration was determined using the bicinchoninic acid protein assay (Sigma-Aldrich).

**Table 1. siRNA oligo sequences.**

|  | Sequence |
|---|---|
| Scrambled siRNA [30] | Sense: 5'-AGCUUCAUAAGGCGCAUGCTT-3' |
|  | Antisense: 5'-GCAUGCGCCUUAUGAAGCUTT-3' |
| NKX3-2 siRNA | Sense: 5'-CCGAGACGCAGGUGAAAAU55-3' |
|  | Antisense: 5'-AUUUUCACCUGCGUCUCGG55-3' |
| RUNX2 siRNA | Sense: 5'- GCACGCUAUUAAAUCCAAA55-3' |
|  | Antisense: 5'-UUUGGAUUUAAUAGCGUGC55-3' |

NKX3-2, RUNX2 and scrambled siRNA sequences are listed.

**Table 2. DNA oligo sequences for RT-qPCR.**

| Gene | Forward primer | Reverse primer |
|---|---|---|
| 28S rRNA | 5'-GCCATGGTAATCCTGCTCAGTAC-3' | 5'-GCTCCTCAGCCAAGCACATAC-3' |
| 18S rRNA | 5'-CGGACCAGAGCGAAAGCA-3' | 5'-ACCTCCGACTTTCGTTCTTGATT-3' |
| 5.8S rRNA | 5'-CACTCGGCTCGTGCGTCGAT-3' | 5'-CGCTCAGACAGGCGTAGCCC-3' |
| Cyclophilin | 5'-CCTGCTTCCACCGGATCAT-3' | 5'-CGTTGTGGCGCGTAAAGTC-3' |
| RUNX2 | 5'-TGATGACACTGCCACCTCTGA-3' | 5'-GCACCTGCCTGGCTCTTCT-3' |
| NKX3-2 | 5'-GCCGCTTCCAAAGACCTAGA-3' | 5'-GCTGCGGTCGCCTGAGA-3' |
| TCOF1 | 5'-AAGCCACCCCAAGACTAGCA -3' | 5'- CCCAGTCTTGCCAGCTTTCT -3' |
| UBTF | 5'-CAGGACCGTGCAGCATATAAAG-3' | 5'-GCCTCGCAGCTTGGTCAT-3' |

Forward and reverse primer sequences are listed for *Homo sapiens*.

### Gene expression analysis

Disruption of cells was performed with TRIzol reagent (Life Technologies; n = 3 samples per condition). RNA isolation and quantification were performed as described previously [5]. cDNA synthesis and real-time quantitative polymerase chain reaction (RT-qPCR) was performed as described previously [23]. Primer sequences are shown in Table 2. Data were analyzed using the standard curve method and RNA (including messenger and non-coding RNA) expression was normalized to cyclophilin as a reference gene. Differential gene expression was determined as fold change relative to control conditions.

The expression of unrelated/unaffected genes which are not controlled by the BMP7-NKX3-2- rRNA axis as a negative control are shown in S1 Fig.

### SUNsET-assay

Translational capacity of SW1353 cells or human primary chondrocytes was assessed with the SUNsET (surface sensing of translation) assay [31, 32]. Cells were incubated for 10 minutes with 10 μg/ml puromycin (Sigma-Aldrich) in the culture medium. After incubation, cells were washed with 0.9% NaCl and disrupted with radioimmunoprecipitation assay (RIPA) buffer. Cell extracts were sonicated on ice at amplitude 10 for 14 cycles (1 s sonication and 1 s pause; Soniprep 150, MSE, Heathfield, UK). DNA concentration was determined using the SYBR-GREEN assay. Equal DNA quantities were loaded on a nitrocellulose membrane by vacuum-pressure. Puromycin labeling was detected using incubation with the primary anti-puromycin antibody 12D10 (Sigma-Aldrich). Bound primary antibodies were detected with rabbit-anti-mouse secondary immunoglobulins, conjugated with horseradish peroxidase (Dako Agilent, Santa Clara, California, USA). Antibodies were visualized by enhanced chemiluminescence (ECL) using the ChemiDoc XRS+ Imaging System (Bio-Rad, Hercules, California, USA). Relative differences were determined as compared to control conditions.

### Statistical analysis

Statistical significance has been calculated with Graphpad PRISM 5.01 (La Jolla, California, USA) using two-tailed paired or unpaired Student's t-tests or one-way analysis of variance (ANOVA) with Bonferroni's Multiple Comparison post-hoc analysis. Details per experiment are indicated in the corresponding figure legends. Significance was set at $p \leq 0.05$ for all tests. To test for normal distribution of the input data, D'Agostino-Pearson omnibus normality tests were performed. All quantitative data sets presented here passed the normality tests. Error bars in graphs represent mean ± standard error of the mean (SEM).

## Results

### BMP7 increases protein translation capacity of SW1353 cells and induces increased rRNA levels

We have previously demonstrated that BMP7 is able to induce a phenotypic switch in OA chondrocytes [23], as well as during chondrogenic differentiation of ATDC5 cells [5]. This change of phenotype is accompanied by synthesis of different proteins that build up the protein-rich part of the articular cartilage ECM [5, 6]. In this study we therefore tested whether BMP7 can influence the protein translational capacity of chondrocytic cells. Treatment of SW1353 cells or human primary chondrocytes with 1 nM BMP7 resulted in a significant increase of overall protein translational capacity compared to the control condition (Fig 1A and 1B). Given the central role of rRNAs in ribosome protein translation [10], we next determined rRNA levels in SW1353 cells or human primary chondrocytes that were exposed to 1 nM BMP7 for 24 hours. In concert with the induced translational capacity following BMP7 treatment, 18S and 5.8S rRNA levels were significantly increased compared to control

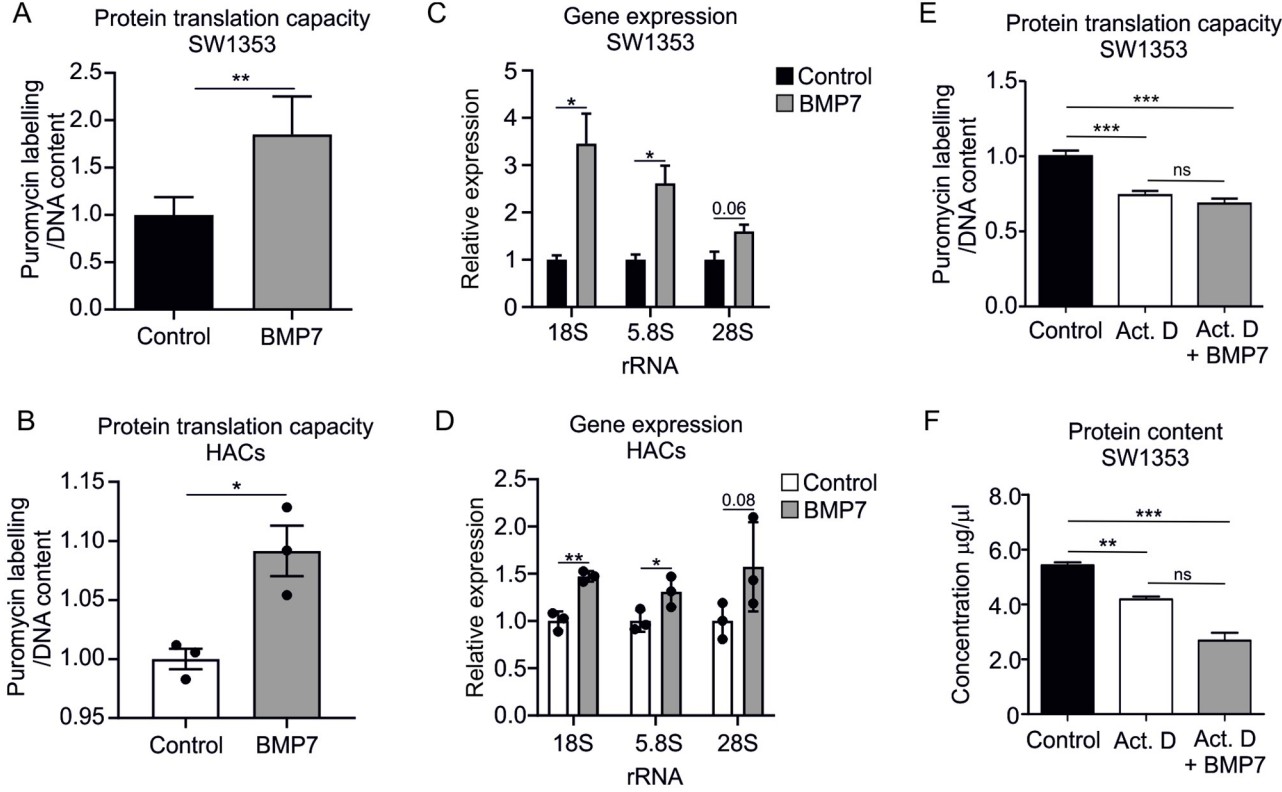

**Fig 1.** BMP7 exposure increases translational capacity of chondrocytic SW1353 cells and is associated with increased rRNA levels Translational capacity was determined using the SUNsET assay in SW1353 cells (**A**) or human primary chondrocytes (n = 3 inidividual donors) (**B**), which were exposed for 24 hours to BMP7 (1nM) or control conditions. Puromycilation data were normalized to DNA content and calculated relative to the control condition (A: n = 6 samples per condition, B: n = 3 samples per donor). In similar samples from A and B, expression levels of 18S rRNA, 5.8S rRNA, and 28S rRNA were determined using RT-qPCR analysis in SW1353 cells (**C**) or human primary chondrocytes (**D**). Data were normalized to cyclophilin expression and set relative to the control condition (C: n = 3 samples per condition, D: n = 3 samples per donor). **E.** Translational capacity was determined in SW1353 cells, which were exposed for 24 hours to Actinomycin D (10 ng/ml) and BMP7 (1 nM) or control conditions. Puromycilation data were normalized to DNA content and calculated relative to the control condition (n = 5 samples per condition). **F.** In similar samples as E, protein content was determined using a BCA assay. Statistical significance was determined using two-tailed unpaired Student's t-tests for A, C, E, F and two-tailed paired Student's t-test for B and D. Bars show the mean ±SEM. * $P<0.05$, ** $P<0.01$, *** $P<0.001$ versus control conditions. ns = not significant.

conditions. 28S rRNA levels were increased, albeit not significant (Fig 1C and 1D) [33]. Protein translation is largely dependent on the transcription of rRNAs [10]. As BMP7 increased protein translational capacity (Fig 1A and 1B), we next asked if active rRNA transcription is involved. To inhibit rRNA transcription, we treated SW1353 cells for 24 hours with 10 ng/ml Actinomycin D (a concentration that selectively inhibits RNA polymerase I [34]), and tested whether the observed Actinomycin D-dependent inhibition of protein translation could be rescued by BMP7. We observed that BMP7 could not rescue the Actinomycin D-mediated inhibition of protein synthesis (Fig 1E and 1F). Collectively, these data demonstrate that BMP7 increases translational capacity in SW1353 cells and regulates rRNA levels.

## BMP7 enhances transcriptional activity of the 47S rDNA promoter reporter

We next determined whether the BMP7-dependent increase of rRNA expression is the result of increased transcription of the 47S rDNA gene. Therefore, SW1353 cells were transfected with a 47S rDNA promoter reporter plasmid and subsequently exposed to 1 nM BMP7 for 24 hours. Treatment with BMP7 resulted in enhanced transcriptional activity of the 47S rDNA promoter reporter compared to the control condition (Fig 2A). In addition, expression of two members of the transcription factor complex involved in the transcription of the 47S precursor rRNA; UBTF and TCOF1 (Treacle Ribosome Biogenesis Factor 1) [35] were increased upon BMP7 treatment in SW1353 cells (Fig 2B) and human primary chondrocytes (Fig 2C). These results demonstrate that the BMP7-dependent increase of rRNA expression is supported by a higher activity of the 47S rDNA promoter activity.

## BMP7-induced rDNA promotor reporter activity, rRNA levels and translational capacity are NKX3-2 dependent

Repression of transcriptional activity of the 47S rDNA gene in higher eukaryotes has, in part, been attributed to RUNX2 [26–28]. Indeed, knockdown of RUNX2 by transient siRNA transfection in SW1353 cells resulted in decreased expression of RUNX2 and its transcriptional target COL10A1, which was accompanied by an increased expression of 18S, 5.8S and 28S rRNA,

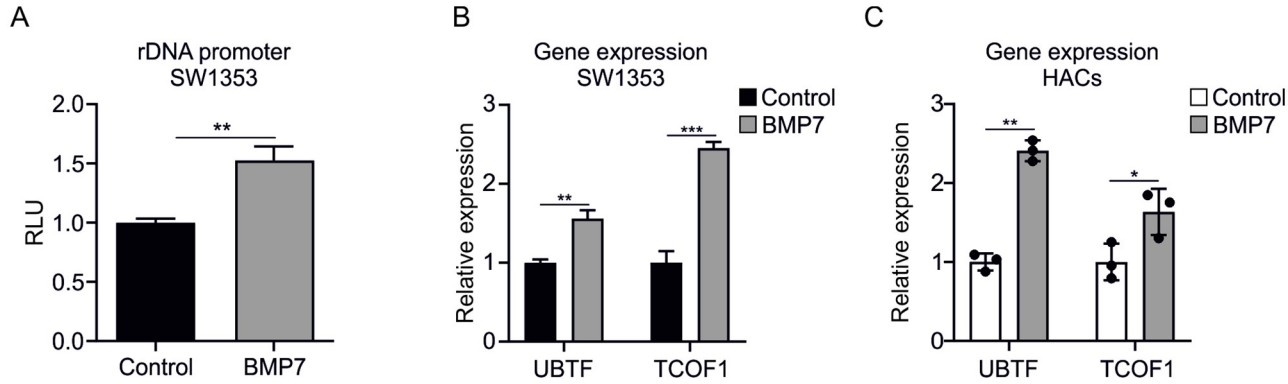

**Fig 2. rDNA promotor reporter activity is increased in SW1353 cells exposed to BMP7.** SW1353 cells were transfected with an NL1.2_47S-rDNA promoter plasmid. Subsequently, cells were exposed to BMP7 (1nM) for 24 hours and Nanoluc luciferase levels were measured (**A**). Data were normalized to DNA content and calculated relative to control conditions (RLU) (n = 6 samples per condition). Expression levels of UBTF and TCOF1 were determined using RT-qPCR analysis in SW1353 cells (**B**) or human primary chondrocytes (**C**). Data were normalized to cyclophilin expression and set relative to the control condition (B: n = 3 samples per condition, C: n = 3 samples per donor). Statistical significance was determined using two-tailed unpaired Student's t-tests for A and B and two-tailed paired Student's t-test for C. Bars show the mean ±SEM. * $P<0.05$, ** $P<0.01$, *** $P<0.001$ versus control conditions.

UBTF and TCOF1 (Fig 3A). Furthermore, RUNX2 knockdown resulted in an increased protein translational capacity (Fig 3B).

In turn, BMP7 is a potent repressor of RUNX2 expression via NKX3-2 [23]. We therefore investigated whether the BMP7-dependent increase of rRNA levels is modulated via NKX3-2. SW1353 cells were transiently transfected with a NKX3-2 siRNA duplex and cells were then incubated in the presence or absence of BMP7. In agreement with our previous findings [23], BMP7 potently induced expression of NKX3-2 (Fig 3C), while the expression of RUNX2 was inhibited (Fig 3D). Knockdown of NKX3-2 could not be counteracted by BMP7 (Fig 3C) and caused a marked increase of RUNX2 expression (Fig 3D). To determine whether the BMP7-dependent increase of rDNA promoter activity is facilitated by NKX3-2, an rDNA promoter reporter assay was conducted with BMP7 under NKX3-2 knockdown conditions. BMP7 increased the transcriptional activity of the rDNA promoter reporter (Fig 3E). NKX3-2 knockdown caused a marked reduction in rDNA promoter reporter activity (Fig 3E). The reduced rDNA promoter activity could not be rescued by stimulation with BMP7. To confirm whether the alterations in rDNA promoter activity result in changes at the rRNA level, 18S, 5.8S and 28S rRNAs, UBTF and TCOF1 were measured. 18S and 5.8S rRNA levels and UBTF and TCOF expression were significantly induced by BMP7 (Fig 3F, cf. Fig 1B). In contrast, the expression levels of 18S, 5.8S and 28S rRNA and UBTF and TCOF expression were reduced in NKX3-2 knockdown conditions (Fig 3F) which was a NKX3-2 knockdown specific reaction (S2 Fig). This could not be restored when NKX3-2 knockdown cells were stimulated with BMP7 (Fig 3F). Similar gene expression responses were detected in human primary chondrocytes and ATDC5 chondrocytes (Fig 3G and S3 Fig). In addition, the increased protein translational capacity under BMP7 treatment (Fig 1A) was also dependent on NKX3-2 expression (Fig 3H).

Overexpression of NKX3-2 by transfecting SW1353 cells or human primary chondrocytes with a validated 3xFLAG-NKX3-2 vector [23] resulted in a significantly decreased expression of RUNX2, which was accompanied by an increased expression of 18S rRNA, and 5.8S rRNA, 28S rRNA, UBTF and TCOF1 (Fig 4A–4C). In addition, translational capacity was also increased in the FLAG-NKX3-2 overexpressing SW1353 cells (Fig 4D). These data are reciprocal to the NKX3-2 knockdown presented in Fig 3 and in line with the BMP7-mediated increase in rRNA expression from Figs 1 and 3. Together these results indicate that the BMP7-induced rDNA promoter activity and translational capacity is NKX3-2-dependent and associated with RUNX2 expression.

## Discussion

The cell's protein translational capacity is directly linked to ribosome biogenesis and highly dependent on the levels of intracellularly available mature rRNAs [11, 12]. Indeed, our data show that the increased translation capacity induced by BMP7 is accompanied by elevated levels of 18S and 5.8S rRNAs (28S non-significantly increased). We found that this is likely caused by a BMP7-dependent induction of rRNA transcription.

Basic activity of rRNA transcription from the rDNA gene has been described to be maintained by signaling through the insulin and insulin-like growth factor axis, via epidermal growth factor, as well as multiple serum components [11, 20–22, 35–37]. These signaling pathways represent the main physiological routes that are at the basis of cellular protein homeostasis during tissue growth and maintenance. Further control of rRNA synthesis occurs through additional control of basic rRNA transcription [38]. Our current study for the first time demonstrates a BMP-mediated control over rRNA transcription in chondrocytic cells. Our previous work showed that BMP7 is a potent morphogen capable of functionally directing the

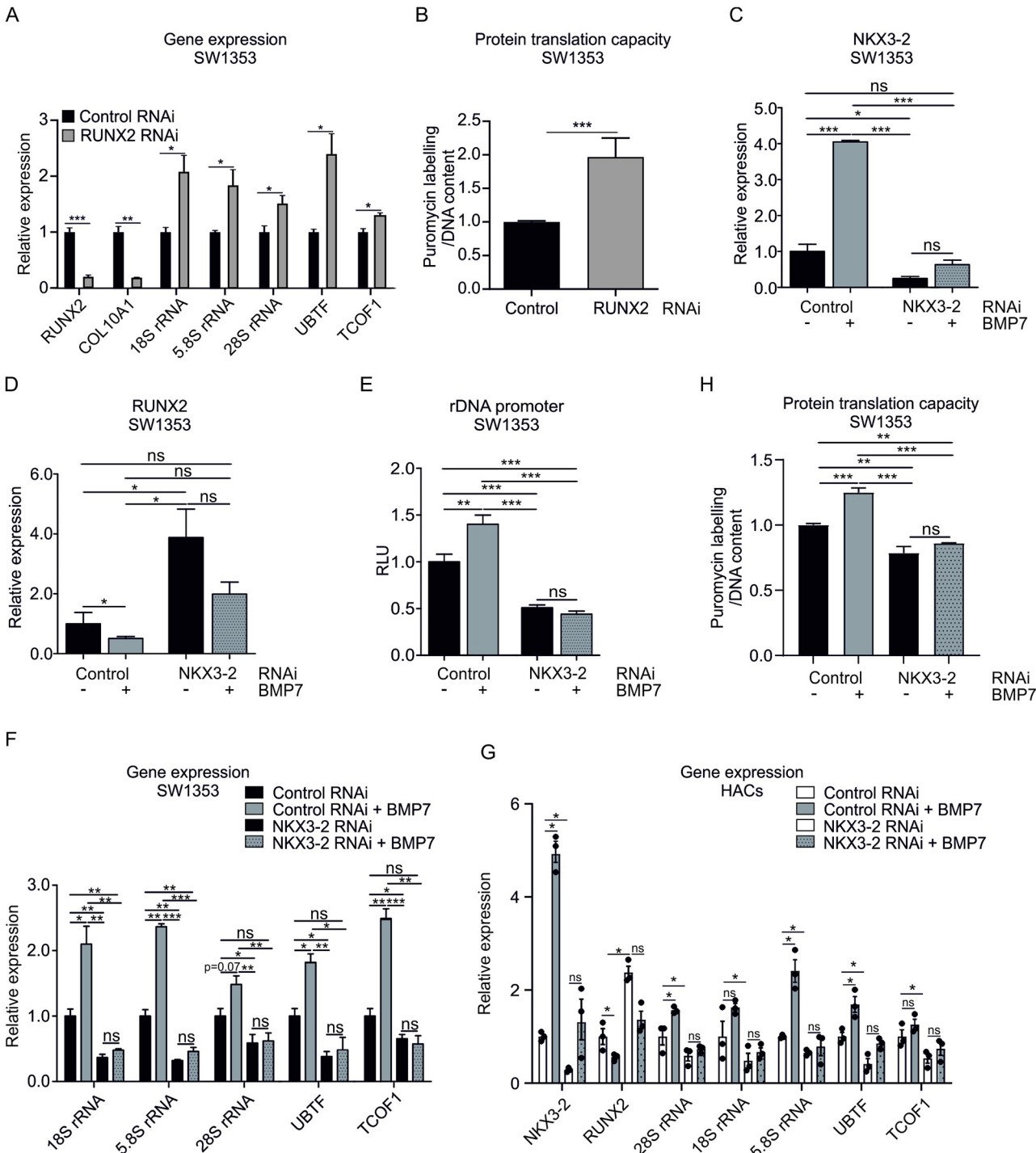

**Fig 3. BMP7-induced promotor reporter activity and rRNA levels are NKX3-2 dependent.** SW1353 cells were transfected with either a scrambled (Control RNAi) or NKX3-2 (NKX3-2 RNAi) siRNA duplex (100nM) and after 24 hours expression levels of RUNX2, COL10A1, 18S rRNA, 5.8S rRNA, 28S rRNA, UBTF and TCOF1 were measured by RT-qPCR. Data were normalized to cyclophilin expression and set relative to the control condition (n = 3 samples per condition) (**A**). In similar samples from A, translational capacity was determined using the SUNsET assay. Puromycilation data were normalized to DNA content and calculated relative to the control condition (n = 6 samples per condition) (**B**). SW1353 cells or human primary chondrocytes were transfected with either a scrambled (Control RNAi) or NKX3-2 (NKX3-2 RNAi) siRNA duplex (100nM) and exposed to BMP7 (1nM) for 24 hours after which expression levels of NKX3-2 (**C,G**) and RUNX2 (**D, G**) were determined using RT-qPCR analysis. Data were normalized to cyclophilin expression and set relative to the SCR control condition (n = 3 samples per condition). **E**. In similar samples as Fig 3C, SW1353 cells were subsequently transfected with a pNL1.2_47S-rDNA promoter plasmid and exposed to BMP7 (1nM) for 24 hours after which

Nanoluc luciferase levels were measured. Data were normalized to DNA content and calculated relative to SCR control conditions (RLU) (n = 6 samples per condition). **F/G**. In similar samples from C and G, expression levels of 18S rRNA, 5.8S rRNA, 28S rRNA, UBTF and TCOF were determined using RT-qPCR analysis. **H**. In similar samples from C, translational capacity was determined using the SUNsET assay. Puromcilation data were normalized to DNA content and calculated relative to the control condition (n = 6 samples per condition) Statistical significance was determined using a 1-way ANOVA with a Bonferroni's Multiple Comparison test (E,H) or two-tailed Student's t-tests (unpaired: A-D, F; paired: G). Bars show the mean ±SEM. * P<0.05, ** P<0.01, *** P<0.001. ns = not significant.

phenotype of developing chondrocytes and of osteoarthritic chondrocytes towards a phenotype that is less hypertrophic [5, 23]. Chondrocyte phenotypic changes are expected to be associated with important changes in the chondrocyte's (extracellular cartilaginous) proteome. Therefore, it remains to be determined what the proteomic consequences are of the

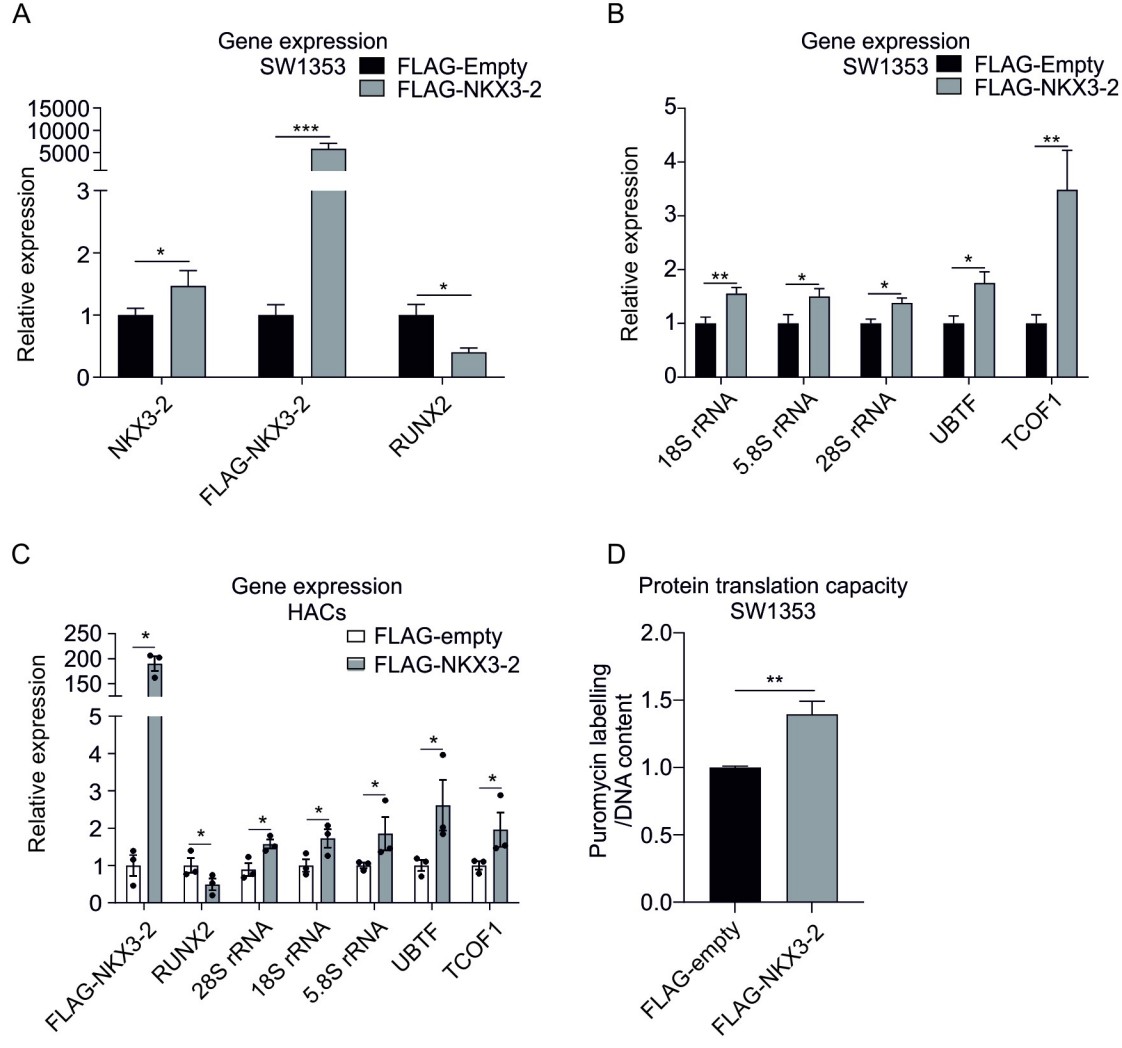

**Fig 4. NKX3-2 overexpression increases rRNA levels and is associated with increased translational capacity.** NKX3-2 was overexpressed by transient transfection of a codon usage optimized FLAG-NKX3-2 vector into SW1353 cells (**A,B**) or primary chondrocytes (n = 3 donors) (**C**). FLAG-empty vector was used as a negative control. Expression of mRNA for the indicated genes was determined by real-time RT-qPCR. Data were normalized to cyclophilin expression and set relative to the control condition (A-B: n = 3 samples per condition, C: n = 3 samples per donor). **D**. Translational capacity was determined in SW1353 cells with and without overexpression of NKX3-2. Puromcilation data were normalized to DNA content and calculated relative to the control condition (n = 5 samples per condition). Statistical significance was determined using two-tailed Student's t-tests (unpaired for: A,B, D; paired for: C). Bars show the mean ±SEM. * P<0.05, ** P<0.01, *** P<0.001 versus control conditions.

BMP7-mediated induction of rRNA transcription and translation, and whether this results in a chondrocyte phenotype change detectable at the protein level.

Downstream of cellular signaling by morphogens, the transcription of the 47S precursor rRNA is driven by a transcription factor complex consisting of SL1 (selective factor 1), UBTF, TIF-IA (transcription initiation factor-IA), TBP (TATA-Box Binding Protein) and RNA Polymerase I [35]. This complex is the basic driver of the RNA polymerase I-dependent transcription of the rDNA gene and forms the basis for modulation of transcription. An important part of the activity of this complex is regulated by the phosphorylation of UBTF, which can be mediated by signaling of ERK, mTOR (mammalian target of rapamycin), CK2 (Casein kinase II), CDK (Cyclin-dependent kinase), CBP (CREB-binding protein) and others [38]. Activities of RUNX-family transcription factors have been shown to control rRNA transcription [39] (Fig 3A). In addition, RUNX2 suppresses rRNA transcription via interacting with UBTF [27] and recruitment of HDAC [26], modulating acetylation of UBF and histones at rDNA loci in osteocytes. Although this mechanism has not yet been described in chondrocytes [40], it does provide an important potential link between BMP7 and the here observed consequence of BMP7 on rRNA transcription. Runx2 is a key transcription factor driving chondrocyte hypertrophy, and the hypertrophy suppressive action of BMP7 on chondrocytes involves the inhibition of Runx2 expression. Importantly, BMP7-mediated inhibition of chondrocyte Runx2 expression has been shown to depend on NKX3-2, a pivotal factor in balancing the hypertrophic differentiation program of chondrocytes and particularly known for its repressive action on *RUNX2* transcription [23, 25]. In this manner, BMP7-dependent NKX3-2 expression is able to control RUNX2 levels, thereby delivering an important contribution to tuning chondrocyte hypertrophy. The current data indicate that NKX3-2 is also involved in the transcription of rRNA, and needed for the here observed BMP7-mediated induction of rRNA transcription. At this point, the precise mode-of-action of the involvement of NKX3-2 in rRNA transcription remains to be determined. However, since NKX3-2 is well-known for its repression of Runx2 expression, we speculate that NKX3-2 is able to regulate rRNA transcription by determining the levels of Runx2, a repressor of rRNA transcription [26–28]. In this manner BMP7 would be able to induce rRNA transcription via attenuation of Runx2 levels. Future Chromatin immunoprecipitation (ChIP)-assays, addressing RUNX2 occupancy on the rDNA gene promotor, should be able to resolve this.

Our data demonstrate the involvement of BMP7 in RNA polymerase I-directed rDNA gene transcription. Ribosome biogenesis is intricately linked to the rate of protein synthesis and mainly controlled at the level of rDNA transcription by RNA polymerase I [11, 12]. However, ribosome biogenesis is a result of a tightly regulated cascade of molecular events, in which processing of the 47S rRNA precursor gives rise to mature 18S, 5.8S and 28S rRNAs. Supporting our present findings on a role for BMP7 in ribosome biogenesis, we previously showed that BMP7 induces the expression of U3 snoRNA [33]. U3 snoRNA is a key factor in the maturation of rRNAs from the 47S rRNA precursor. This indicates that BMP7 is able to orchestrate rRNA transcription with its maturation. This notion also has implications for BMP7 and NKX3-2 as chondrocyte phenotypic modulators and highlights that morphogen-mediated cellular differentiation is orchestrated with ribosome biogenesis via key transcription factors that are centrally involved in determining the cellular phenotype. Indeed, a similar link between PTHrP (Parathyroid hormone-related peptide; a well-known chondrocyte phenotype regulating morphogen [41]) and chondrocyte ribosome biogenesis was previously reported [42].

Taken together, the results of this study demonstrate that BMP7 increases protein translation and promotes rRNA transcription in SW1353 cell and human primary chondrocytes via an NKX3-2-dependent route (as schematically depicted in Fig 5). It is well established that BMP7 controls the expression of key chondrocyte transcription factors that direct the

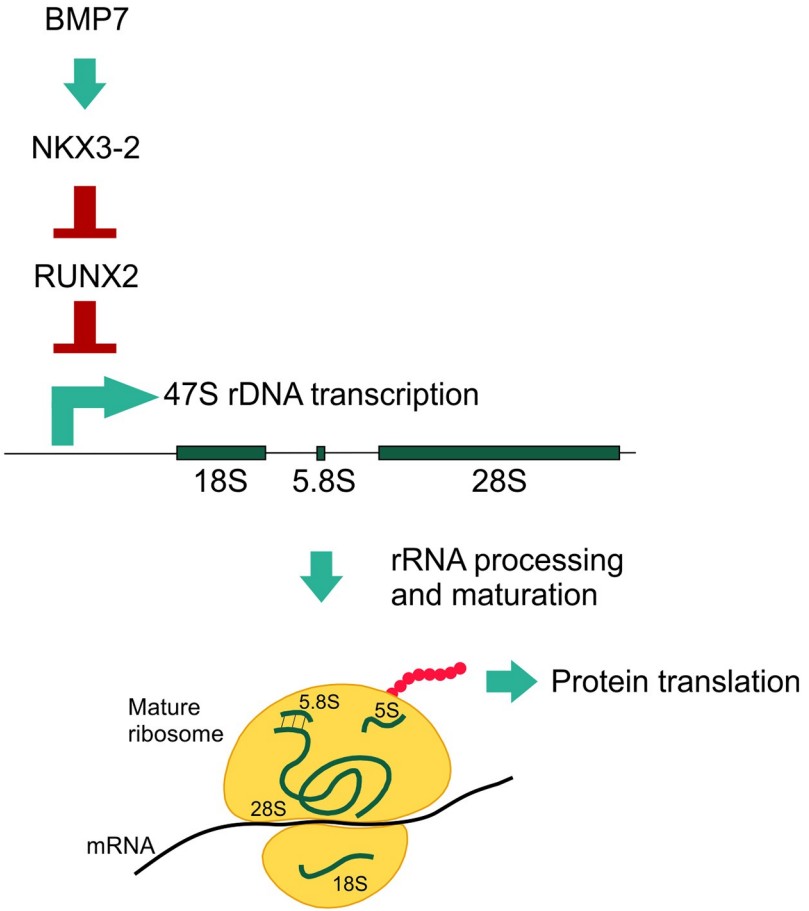

**Fig 5. Graphical representation of the elucidated mechanism.** The results of this study demonstrate that BMP7 increases protein translation and promotes rRNA transcription in SW1353 cells. rRNA transcription in SW1353 cells has been demonstrated to be mediated by a BMP7-induced inhibition of RUNX2 in a NKX3-2-dependent manner.

chondrocyte's phenotype. Our results now additionally uncover that this is associated with changes in the chondrocyte's capacity to synthesize ribosomes. Since BMP7 is being explored as a potential treatment option for OA [43–45], this provides a broader understanding of the spectrum of the mode-of-action of BMP7 as an OA disease-modifying molecule.

## Supporting information

**S1 Fig. Unrelated/unaffected genes which are not controlled by the BMP7-NKX3-2- rRNA axis. A/B.** SW1353 cells were transfected with either a scrambled (SCR) or NKX3-2 (NKX3-2 KD) siRNA duplex (100nM) and exposed to BMP7 (1nM) for 24 hours after which expression levels of Viperin, NFAT5, KRT18 or PTCH1 were determined using RT-qPCR analysis. Data were normalized to cyclophilin expression and set relative to the SCR control condition (n = 3 samples per condition). Statistical significance was determined using a two-tailed unpaired Student's t-tests, an no significant changes for each gene between conditions was observed. Bars show the mean ±SEM. * $P<0.05$, ** $P<0.01$, *** $P<0.001$.
(TIF)

**S2 Fig. BMP7-induced rRNA levels are NKX3-2 dependent in ATDC5 cells.** ATDC5 cells
(6.400 cells/cm$^2$) were differentiated in the chondrogenic lineage for 6 days to acquire a chondrocyte phenotype and then transfected (according to the manufacturer's protocol, using HiPerfect, Qiagen) with either a scrambled (Control RNAi (Eurogentec)) or Nkx3-2 (Nkx3-2 RNAi) siRNA duplex (100nM; sense: 5′-CAGAGACGCAAGUGAAGAUTT-3′, anti-sense: 5′-AUCUUCACUUGCGUCUCUGTT-3′) and exposed to BMP7 (1nM) for 24 hours after which expression levels of Nkx3-2, Runx2, 18S rRNA, 5.8S rRNA, 28S rRNA, Ubtf and Tcof1 were measured by RT-qPCR. Data were normalized to cyclophilin expression and set relative to the control condition (n = 3 samples per condition). Differentiation medium for ATDC5 consisted of Dulbecco's minimal essential medium (DMEM)/F12 (Invitrogen), 5% fetal calf serum (FCS) (Sigma-Aldrich), 1% antibiotic/antimycotic (Invitrogen) and 1% NEAA (non-essential amino acids; Invitrogen), 10mg/ml insulin (Sigma-Aldrich), 10mg/ml transferrin (Roche) and 30 nM sodium selenite (Sigma-Aldrich). Statistical significance was determined using a two-tailed unpaired Student's t-tests. Bars show the mean ±SEM. * P<0.05, ** P<0.01, *** P<0.001. ns = not significant.
(TIF)

**S3 Fig. NKX3-2 overexpression increases rRNA levels and is associated with increased translational capacity. A-B**. NKX3-2 was overexpressed by transient transfection of a codon usage optimized FLAG-NKX3-2 vector into SW1353 cells and FLAG-empty vector was used as a negative control. The next day, these cells were transfected with either a scrambled (Control RNAi) or NKX3-2 (NKX3-2 RNAi) siRNA duplex (100nM) and after 24 hours cells were harvested. Expression of mRNA for the indicated genes was determined by real-time RT-qPCR. Data were normalized to cyclophilin expression and set relative to the control condition (A-B: n = 3 samples per condition). **C**. Translational capacity was determined and Puromycilation data were normalized to DNA content and calculated relative to the control condition (n = 5 samples per condition). Statistical significance was determined using unpaired two-tailed Student's t-tests. Bars show the mean ±SEM. * P<0.05, ** P<0.01, *** P<0.001 versus control conditions. ns = not significant.
(TIF)

**S1 File.**
(PDF)

## Author Contributions

**Conceptualization:** Ellen G. J. Ripmeester, Tim J. M. Welting, Guus G. H. van den Akker, Don A. M. Surtel, Andy Cremers, Lodewijk W. van Rhijn, Marjolein M. J. Caron.

**Data curation:** Ellen G. J. Ripmeester, Tim J. M. Welting, Guus G. H. van den Akker, Don A. M. Surtel, Jessica S. J. Steijns, Andy Cremers, Marjolein M. J. Caron.

**Formal analysis:** Ellen G. J. Ripmeester, Guus G. H. van den Akker, Don A. M. Surtel, Jessica S. J. Steijns, Andy Cremers, Marjolein M. J. Caron.

**Funding acquisition:** Tim J. M. Welting, Lodewijk W. van Rhijn.

**Investigation:** Ellen G. J. Ripmeester, Tim J. M. Welting, Guus G. H. van den Akker, Jessica S. J. Steijns, Andy Cremers, Lodewijk W. van Rhijn, Marjolein M. J. Caron.

**Methodology:** Ellen G. J. Ripmeester, Tim J. M. Welting, Guus G. H. van den Akker, Don A. M. Surtel, Jessica S. J. Steijns, Andy Cremers, Marjolein M. J. Caron.

**Project administration:** Tim J. M. Welting, Guus G. H. van den Akker, Don A. M. Surtel, Jessica S. J. Steijns, Andy Cremers, Lodewijk W. van Rhijn, Marjolein M. J. Caron.

**Resources:** Tim J. M. Welting, Don A. M. Surtel, Andy Cremers, Lodewijk W. van Rhijn.

**Supervision:** Tim J. M. Welting, Guus G. H. van den Akker, Lodewijk W. van Rhijn, Marjolein M. J. Caron.

**Validation:** Ellen G. J. Ripmeester, Guus G. H. van den Akker, Lodewijk W. van Rhijn, Marjolein M. J. Caron.

**Visualization:** Jessica S. J. Steijns, Marjolein M. J. Caron.

**Writing – original draft:** Ellen G. J. Ripmeester, Tim J. M. Welting, Guus G. H. van den Akker, Jessica S. J. Steijns, Marjolein M. J. Caron.

**Writing – review & editing:** Ellen G. J. Ripmeester, Tim J. M. Welting, Guus G. H. van den Akker, Don A. M. Surtel, Jessica S. J. Steijns, Andy Cremers, Lodewijk W. van Rhijn, Marjolein M. J. Caron.

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
