## [Decision Letter · Decision Letter 0]

25 Jan 2021

PONE-D-20-37112

BMP7 increases protein synthesis in SW1353 cells and determines rRNA levels in a NKX3-2-dependent manner

PLOS ONE

Dear Dr. Caron,

Thank you for submitting your manuscript to PLOS ONE. After careful consideration, we feel that it has merit but does not fully meet PLOS ONE’s publication criteria as it currently stands. Therefore, we invite you to submit a revised version of the manuscript that addresses the points raised during the review process.

We regret that it took a long time to evaluate your work, but it is not for lack of editorial effort. We consulted more than 30 experts (including suggested referees), and only recently we secured a second opinion. One of the reviewers is rather critical of your work, and we recommend that you address these comments to the extend possible by experimentation or otherwise modifications in the text. In the latter case, unresolvable concerns by reviewers as always could be adopted by you as your own acknowledgment of limitations to your work in the Discussion section. You could then take the opportunity to add how your work now adds to the field regardless of the current limitations, similar to the statements you would otherwise make in the rebuttal letter. The second reviewer was more enthusiastic about your work and provided constructive recommendations for revision that should not be insurmountable.   

We look forward to receiving your revised manuscript.

Kind regards,

Andre van Wijnen

Academic Editor

PLOS ONE

Journal Requirements:

2.We note that you have a patent relating to material pertinent to this article. Please provide an amended statement of Competing Interests to declare this patent (with details including name and number), along with any other relevant declarations relating to employment, consultancy, patents, products in development or modified products etc. Please confirm that this does not alter your adherence to all PLOS ONE policies on sharing data and materials, as detailed online in our guide for authors http://journals.plos.org/plosone/s/competing-interests by including the following statement: "This does not alter our adherence to  PLOS ONE policies on sharing data and materials.” If there are restrictions on sharing of data and/or materials, please state these. Please note that we cannot proceed with consideration of your article until this information has been declared.

Reviewers' comments:

Reviewer's Responses to Questions

**Comments to the Author**

1. Is the manuscript technically sound, and do the data support the conclusions?

Reviewer #1: Yes

Reviewer #2: Partly

2. Has the statistical analysis been performed appropriately and rigorously? 

Reviewer #1: Yes

Reviewer #2: Yes

3. Have the authors made all data underlying the findings in their manuscript fully available?

Reviewer #1: Yes

Reviewer #2: Yes

4. Is the manuscript presented in an intelligible fashion and written in standard English?

Reviewer #1: Yes

Reviewer #2: Yes

5. Review Comments to the Author

Reviewer #1: The authors have already published that BMP7 regulates rRNA levels via NKX3.2. Thus, the first two figures in this manuscript replicate their own previously published data, but using SW1353 cells rather than primary HACs. In addition, previously published studies have established a role for NKX3.2-mediated repression of RUNX2 in chondrogenic differentiation. Thus, this work fails to meet the journals requirement for novelty.

Additional issues:

Major: The siRNA studies in Figure 3 lack necessary specificity controls. These include; (1) the failure to use more than one NKX3-2 targeted siRNA duplex (2-3 distinct siRNAs should be used to reduce chance of off-target effects), (2) The authors should include a rescue experiment, by re-expression of a NKX3-2 mutant that evades riRNA silencing.

Reviewer #2: The authors present a nicely written manuscript which investigates an interesting aspect in chondrogenic cell biology which not too many labs enquire. However, to fully prove the hypothesis presented, this work needs some further extension.

1. SUNsET assay: it is not very clear how this assay works in SW1353 cells. Please give some more details on how this assay reflects protein translation.

2. To further prove the point of the paper it would be appropriate to show the expression of an unrelated/unaffected gene which is not controlled by the BMP7-NKX3-2- rRNA axis as a negative control.

3. Next, the authors hypothesize that BMP7 influences the protein translational capacity in differentiating chondrocytes leading to a phenotypic change characterized by synthesis of the protein-rich part of the articular cartilage’s ECM. The authors need to demonstrate the direct link between the increased translation of proteins via ribosomal stimulation by BMP7 and to the articular cartilage ECM . Direct comparison between mRNA expression, protein expression and if possible protein ubiquitination of genes involved in cartilage ECM is necessary.

4. A little graphical representation of the elucidated mechanism/pathway would add value to the manuscript and help all the readers which are not too familiar with control of rRNA's transcription and ribosomal translation.

6. PLOS authors have the option to publish the peer review history of their article (what does this mean?). If published, this will include your full peer review and any attached files.

Reviewer #1: No

Reviewer #2: No

---

## [Author Response · Author response to Decision Letter 0]

8 Mar 2021

Dear Editorial Board Member, Dear Academic Editor Andre van Wijnen,

On behalf of all authors, we are pleased to submit the new revision of the attached original research article entitled: “BMP7 increases protein synthesis in SW1353 cells and determines rRNA levels in a NKX3-2-dependent manner” with manuscript ID PONE-D-20-37112 for publication in PLOS ONE. 

The manuscript has been adjusted according to the Academic Editor and Reviewers’ comments. In the attached files you can find the requested systematic point-by-point description of how we dealt with the Academic Editor and Reviewers comments in the rebuttal letter, as well as a final marked-up and clean unmarked copy of the manuscript and a new figure. 

We again thank the Academic Editor and the Reviewers for their time and effort to critically review our manuscript and hope that our revised manuscript now meets the Academic Editor and Reviewers’ remarks and meets the high standards for publication in PLOS ONE.

All co-authors have read and approved the submitted version of the manuscript. None of the material from this manuscript has been published or is under consideration elsewhere, including the internet. 

Conflicts of Interest: MMJ Caron and TJM Welting are inventors on patents WO2017178251 and WO2017178253 (Owned by Chondropeptix). LW van Rhijn and TJM Welting are shareholder in Chondropeptix and are CDO, and CSO of Chondropeptix, respectively. All other authors have no competing interests to declare. This does not alter our adherence to PLOS ONE policies on sharing data and materials.

On behalf of all co-authors, 

Sincerely,

Marjolein M.J. Caron, PhD

Reviewers' comments:

Reviewer's Responses to Questions

Comments to the Author

1. Is the manuscript technically sound, and do the data support the conclusions?

Reviewer #1: Yes

Reviewer #2: Partly

2. Has the statistical analysis been performed appropriately and rigorously? 

Reviewer #1: Yes

Reviewer #2: Yes

3. Have the authors made all data underlying the findings in their manuscript fully available?

Reviewer #1: Yes

Reviewer #2: Yes

4. Is the manuscript presented in an intelligible fashion and written in standard English?

Reviewer #1: Yes

Reviewer #2: Yes

 

5. Review Comments to the Author

Reviewer #1: The authors have already published that BMP7 regulates rRNA levels via NKX3.2. Thus, the first two figures in this manuscript replicate their own previously published data, but using SW1353 cells rather than primary HACs. In addition, previously published studies have established a role for NKX3.2-mediated repression of RUNX2 in chondrogenic differentiation. Thus, this work fails to meet the journals requirement for novelty.

• Authors’ response: We thank the Reviewer for carefully reading our manuscript and we will address the remaining issues point-by-point below.

• The Reviewer is correct in stating that Figure 1A and 1B convey the same message as Figures published in the paper by Ripmeester et al. [1]. Furthermore, the link between BMP7-NKX3-2-RUNX2 repression axis was also previously reported by our group [2]. We also previously found that BMP7 can influence U3 snoRNA expression and thereby maturation of the 47S pre-rRNA [1]. However the involvement of BMP7 in 47S pre-rRNA transcription has not been shown before by us or other authors. In addition, the involvement of NKX3-2-RUNX2 repression as a potential mechanism of action for the observed BMP7 mediated increase in 47S pre-rRNA transcription was also not reported before. Indeed, this work builds upon our previous work but reports novel insights into the molecular interaction between BMP7 and chondrocyte ribosome biogenesis.

Additional issues:

Major: The siRNA studies in Figure 3 lack necessary specificity controls. These include; (1) the failure to use more than one NKX3-2 targeted siRNA duplex (2-3 distinct siRNAs should be used to reduce chance of off-target effects), (2) The authors should include a rescue experiment, by re-expression of a NKX3-2 mutant that evades riRNA silencing.

• Authors’ response: The siRNA against NKX3-2 in this manuscript is the same as the siRNA we previously published [2], were we showed more confirmation that knockdown of NKX3-2 by this siRNA resulted in specific effects on NKX3-2 target genes. 

Due to the COVID19 crisis and subsequent restrictions in lab activities, we were not in the possibility to initiate new cell culture experiments plus all downstream activities that would be required for additional siRNA and NKX3-2 re-expression studies. We do however have a previously conducted experiment from which the data further harness that NKX3-2 is involved in regulating the expression of rRNAs. Overexpression of BAPX‐1/NKX3-2 by transfecting SW1353 cells with a validated 3xFLAG‐BAPX‐1/NKX3-2 vector [2] resulted in a significantly decreased expression of RUNX2 accompanied by an increased expression for 18S rRNA and 5.8S rRNA. These data are reciprocal to the NKX3-2 knockdown presented in Figure 3 and in line with the BMP7 mediated increase in rRNA expression from Figure 1. We hope that these data provide additional insight for this Reviewer on the role of NKX3-2 in chondrocyte rRNA expression. This Figure is now included in the Supplemental files accompanying this manuscript. 

S2 Fig: Overexpression of NKX3-2 results in increased 18S rRNA and 5.8S rRNA expression. Overexpression of NKX3-2 was achieved by transfecting SW1353 cells with a validated 3xFLAG-NKX3-2 vector [2]. After 24 hours gene expression analysis was performed using RT-qPCR analysis. Data were normalized to cyclophilin expression and set relative to the empty-FLAG control condition (n=3 samples per condition). Increased FLAG-NKX3-2 expression was determined. Overexpression of FLAG-NKX3-2 resulted in decreased gene expression of RUNX2 and increased expression of 18S rRNA and 5.8S rRNA. 28S rRNA expression was not significantly changed by FLAG-NKX3-2. Statistical significance was determined using a two-tailed unpaired Student’s t-tests. Bars show the mean ±SEM. * P<0.05, ** P<0.01, *** P<0.001. 

Reviewer #2: The authors present a nicely written manuscript which investigates an interesting aspect in chondrogenic cell biology which not too many labs enquire. However, to fully prove the hypothesis presented, this work needs some further extension.

• Authors’ response: We thank the Reviewer for carefully reading our manuscript and emphasizing that research on protein translation in chondrocyte cell biology is not performed by too many labs. We are grateful for his/her positive remarks and we will happily address the remaining issues point-by-point below.

1. SUNsET assay: it is not very clear how this assay works in SW1353 cells. Please give some more details on how this assay reflects protein translation.

• Authors’ response: The SUNsET assay (surface sensing of translation) uses antibody detection of puromycin to monitor translation via immunohistochemistry. Puromycin mimics the terminus of aminoacyl-tRNA and hence is incorporated co-translationally in the growing nascent polypeptide chain. The amount of puromycin-incorporated peptides reflects the protein translational capacity of the cell, which can be determined by the intensity of the fluorescent signal [3].

2. To further prove the point of the paper it would be appropriate to show the expression of an unrelated/unaffected gene which is not controlled by the BMP7-NKX3-2- rRNA axis as a negative control.

• Authors’ response: We agree with the Reviewer that showing unrelated/unaffected gene expression by the BMP7-NKX3-2-rRNA axis would benefit the data presented in the manuscript as a negative control. The expression of the genes below are not significantly affected by either BMP7 or NKX3-2 knockdown, and this Figure is now included in the Supplemental files accompanying this manuscript. 

S1 Fig: Unrelated/unaffected genes which are not controlled by the BMP7-NKX3-2- rRNA axis. A/B. SW1353 cells were transfected with either a scrambled (SCR) or NKX3-2 (NKX3-2 KD) siRNA duplex (100nM) and exposed to BMP7 (1nM) for 24 hours after which expression levels of Viperin, NFAT5, KRT18 or PTCH1 were determined using RT-qPCR analysis. Data were normalized to cyclophilin expression and set relative to the SCR control condition (n=3 samples per condition). Statistical significance was determined using a two-tailed unpaired Student’s t-tests, an no significant changes for each gene between conditions was observed. Bars show the mean ±SEM. * P<0.05, ** P<0.01, *** P<0.001.

3. Next, the authors hypothesize that BMP7 influences the protein translational capacity in differentiating chondrocytes leading to a phenotypic change characterized by synthesis of the protein-rich part of the articular cartilage’s ECM. The authors need to demonstrate the direct link between the increased translation of proteins via ribosomal stimulation by BMP7 and to the articular cartilage ECM. Direct comparison between mRNA expression, protein expression and if possible protein ubiquitination of genes involved in cartilage ECM is necessary.

• Authors’ response: The Reviewer raises a valid point. In order to demonstrate that the effect of BMP7 on increased chondrocyte protein translation depends on rRNA transcription, we conducted an additional experiment. We treated SW1353 cells for 24 hours with Actinomycin D at 10 ng/ml, a concentration that selectively inhibits RNA polymerase I (PMID 21922053) [4]. RNA polymerase I is the only polymerase responsible for 47S rDNA transcription. 

The data demonstrate that Actinomycin D mediated inhibition of rRNA transcription results in reduced chondrocyte protein synthesis. This could not be rescued by BMP7. These data strongly suggest that the effect of BMP7 on chondrocyte protein translation requires active rRNA transcription supporting our hypothesis. These data are now added to the main manuscript as Figure 1 C and D. 

While we fully acknowledge the added value of measuring cartilage ECM specific protein synthesis in this context, we regret that we were not able to meet this request due to COVID19 related restrictions in local lab activities. We hope the Reviewer accepts this force majeure situation. 

A: Translational capacity was determined using the SUNsET assay in SW1353 cells, which were exposed for 24 hours to Actinomycin D (10ng/ml) and BMP7 (1nM) or control conditions. Puromycilation data were normalized to DNA content and calculated relative to the control condition (n=5 samples per condition). Treatment with Actinomycin D reduced translational capacity of SW1353 cells, which could not be rescued by simultaneous exposure to BMP7. B: In similar samples as A, protein content was determined using a BCA assay. Treatment with Actinomycin D reduced total protein content, which could not be rescued by exposure to BMP7. Statistical significance was determined using unpaired Student’s t-tests. Bars show the mean (±SEM). ** P<0.01, *** P<0.001.

4. A little graphical representation of the elucidated mechanism/pathway would add value to the manuscript and help all the readers which are not too familiar with control of rRNA's transcription and ribosomal translation.

• Authors’ response: We agree with the Reviewer that a graphical representation of the elucidated mechanism would benefit the manuscript. We thank the Reviewer for this helpful suggestion and included the graphical representation as shown below in the manuscript as Figure 4. 

Fig 4: Graphical representation of the elucidated mechanism. The results of this study demonstrate that BMP7 increases protein translation and promotes rRNA transcription in SW1353 cells. rRNA transcription in SW1353 cells has been demonstrated to be mediated by a BMP7-induced inhibition of RUNX2 in a NKX3-2-dependent manner. 

References: 

1. Ripmeester EGJ, Caron MMJ, van den Akker GGH, Surtel DAM, Cremers A, Balaskas P, et al. Impaired chondrocyte U3 snoRNA expression in osteoarthritis impacts the chondrocyte protein translation apparatus. Scientific reports. 2020;10(1):13426-. doi: 10.1038/s41598-020-70453-9. PubMed PMID: 32778764.

2. Caron MMJ, Emans PJ, Surtel DAM, van der Kraan PM, van Rhijn LW, Welting TJM. BAPX-1/NKX-3.2 Acts as a Chondrocyte Hypertrophy Molecular Switch in Osteoarthritis. Arthritis & Rheumatology. 2015;67(11):2944-56. doi: 10.1002/art.39293.

3. Goodman CA, Hornberger TA. Measuring protein synthesis with SUnSET: a valid alternative to traditional techniques? Exerc Sport Sci Rev. 2013;41(2):107-15. Epub 2012/10/24. doi: 10.1097/JES.0b013e3182798a95. PubMed PMID: 23089927; PubMed Central PMCID: PMCPMC3951011.

4. Bensaude O. Inhibiting eukaryotic transcription: Which compound to choose? How to evaluate its activity? Transcription. 2011;2(3):103-8. Epub 2011/09/17. doi: 10.4161/trns.2.3.16172. PubMed PMID: 21922053; PubMed Central PMCID: PMCPMC3173647.

---

## [Decision Letter · Decision Letter 1]

29 Mar 2021

PONE-D-20-37112R1

BMP7 increases protein synthesis in SW1353 cells and determines rRNA levels in a NKX3-2-dependent manner

PLOS ONE

Dear Dr. Caron,

Thank you for submitting your revised manuscript to PLOS ONE. After careful consideration, we feel that it has merit but does not fully meet PLOS ONE’s publication criteria as it currently stands. Therefore, we invite you to submit a revised version of the manuscript that addresses the points raised during the review process.

One of the reviewers expressed residual concerns that some of the additional work that was recommended, while logistically very difficult under COVID conditions, would still add value to this work. From an editorial perspective, we have rendered an initial decision based on the reviews of only one reviewer to expedite a decision in light of the fact that many suitable reviewers were not available to assess your work. Since this reduced the number of points that needed to be addressed, it is perhaps not unreasonable for you to reconsider what could be done to satisfy the recommendations of this single reviewer.

Upon personal review of this work, I appreciate the importance of your work. However, it is objectively evident that your paper, even with the supplementary figures you provided, contains a rather minimal amount of data documented in Figures 1 and 3, beyond a simple two component bar graph in Figure 2. Papers for PLOS One typically present a larger volume of high quality work. 

Moving forward, I recommend that you follow the advice of the reviewer and let this paper mature and include a bit more data at an extended deadline to allow you to address the original critique. In addition, I believe that the Supplementary Figures should be incorporated in the main text because they present important context for the main findings. With the number of figures even in the revised paper you are not reaching a maximal figure limit yet. We trust that you will be able to make additional modifications, but feel free to contact us by email if you have questions regarding what you could realistically do and not do with an extended deadline. 

We look forward to receiving your revised manuscript.

Kind regards,

Andre van Wijnen

Academic Editor

PLOS ONE

Reviewers' comments:

Reviewer's Responses to Questions

**Comments to the Author**

1. If the authors have adequately addressed your comments raised in a previous round of review and you feel that this manuscript is now acceptable for publication, you may indicate that here to bypass the “Comments to the Author” section, enter your conflict of interest statement in the “Confidential to Editor” section, and submit your "Accept" recommendation.

Reviewer #1: (No Response)

2. Is the manuscript technically sound, and do the data support the conclusions?

Reviewer #1: Partly

3. Has the statistical analysis been performed appropriately and rigorously? 

Reviewer #1: Yes

4. Have the authors made all data underlying the findings in their manuscript fully available?

Reviewer #1: Yes

5. Is the manuscript presented in an intelligible fashion and written in standard English?

Reviewer #1: Yes

6. Review Comments to the Author

Reviewer #1: The authors have responded in a fashion that IF they carried through to complete the studies requested might address earlier critiques. However, due to COVID19 related research restrictions they are unable to complete the necessary experiments. While this is regrettable, the inability to generate this data does not reduce its importance. I would suggest that the authors be permitted an extension until such a time as research activity is restored and these studies can be completed.

7. PLOS authors have the option to publish the peer review history of their article (what does this mean?). If published, this will include your full peer review and any attached files.

Reviewer #1: No

---

## [Author Response · Author response to Decision Letter 1]

15 Nov 2021

PONE-D-20-37112R1

BMP7 increases protein synthesis in SW1353 cells and determines rRNA levels in a NKX3-2-dependent manner

PLOS ONE

Dear Dr. Caron,

Thank you for submitting your revised manuscript to PLOS ONE. After careful consideration, we feel that it has merit but does not fully meet PLOS ONE’s publication criteria as it currently stands. Therefore, we invite you to submit a revised version of the manuscript that addresses the points raised during the review process.

One of the reviewers expressed residual concerns that some of the additional work that was recommended, while logistically very difficult under COVID conditions, would still add value to this work. From an editorial perspective, we have rendered an initial decision based on the reviews of only one reviewer to expedite a decision in light of the fact that many suitable reviewers were not available to assess your work. Since this reduced the number of points that needed to be addressed, it is perhaps not unreasonable for you to reconsider what could be done to satisfy the recommendations of this single reviewer.

Upon personal review of this work, I appreciate the importance of your work. However, it is objectively evident that your paper, even with the supplementary figures you provided, contains a rather minimal amount of data documented in Figures 1 and 3, beyond a simple two component bar graph in Figure 2. Papers for PLOS One typically present a larger volume of high quality work. 

Moving forward, I recommend that you follow the advice of the reviewer and let this paper mature and include a bit more data at an extended deadline to allow you to address the original critique. In addition, I believe that the Supplementary Figures should be incorporated in the main text because they present important context for the main findings. With the number of figures even in the revised paper you are not reaching a maximal figure limit yet. We trust that you will be able to make additional modifications, but feel free to contact us by email if you have questions regarding what you could realistically do and not do with an extended deadline. 

We look forward to receiving your revised manuscript.

Kind regards,

Andre van Wijnen

Academic Editor

PLOS ONE

• Authors’ response: We thank the Academic Editor for carefully rereading our manuscript and the generous opportunity and time to revise our manuscript based on comments from the Academic Editor and the Reviewer. 

• To meet the Editors’ request regarding the rather minimal amount of data presented in the Figures, we repeated the majority of the experiments in human primary chondrocytes (3 independent donors) and obtained similar results. These data are now added to the manuscript (Figure 1B, 1D, 2C, 3G and 4C). Moreover, we moved the NKX3-2 overexpression gene expression data that was presented in S2 Fig to the main manuscript (Figure 4), repeated the experiment in human primary chondrocytes and added translational capacity data for SW1353 cells. In addition to the presented rRNA promoter data presented in Figure 2, we now measured gene expression of UBTF and TCOF1, two factors critically involved in the transcription of the 47S precursor rRNA (Figure 2, 3 and 4). We find that this additional work adds value to the original presented work and now adheres to the volume and quality of work for publication in PLOS One. 

Comments to the Author

1. If the authors have adequately addressed your comments raised in a previous round of review and you feel that this manuscript is now acceptable for publication, you may indicate that here to bypass the “Comments to the Author” section, enter your conflict of interest statement in the “Confidential to Editor” section, and submit your "Accept" recommendation.

Reviewer #1: (No Response)

2. Is the manuscript technically sound, and do the data support the conclusions?

Reviewer #1: Partly

3. Has the statistical analysis been performed appropriately and rigorously? 

Reviewer #1: Yes

4. Have the authors made all data underlying the findings in their manuscript fully available?

Reviewer #1: Yes

5. Is the manuscript presented in an intelligible fashion and written in standard English?

Reviewer #1: Yes

6. Review Comments to the Author

Reviewer #1: The authors have responded in a fashion that IF they carried through to complete the studies requested might address earlier critiques. However, due to COVID19 related research restrictions they are unable to complete the necessary experiments. While this is regrettable, the inability to generate this data does not reduce its importance. I would suggest that the authors be permitted an extension until such a time as research activity is restored and these studies can be completed.

• Authors’ response: We thank the Reviewer for carefully rereading our manuscript and we will address the remaining issue below. 

The Reviewers previous comment to our manuscript that we could not address due to COVID19 was: Major: The siRNA studies in Figure 3 lack necessary specificity controls. These include; (1) the failure to use more than one NKX3-2 targeted siRNA duplex (2-3 distinct siRNAs should be used to reduce chance of off-target effects), (2) The authors should include a rescue experiment, by re-expression of a NKX3-2 mutant that evades riRNA silencing.

• Authors’ response: The siRNA against NKX3-2 in this manuscript is the same as the siRNA we previously published [1], were we showed more confirmation that knockdown of NKX3-2 by this siRNA resulted in specific effects on NKX3-2 target genes. We now ordered and tested two new siRNA duplexes for human NKX3-2, but these did not result in successful knockdown of NKX3-2. As a second approach to try to answer the Reviewers request, we designed an siRNA duplex for mouse NKX3-2 and obtained a successful knockdown and similar results regarding rRNA expression in the murine ATDC5 cell line that was pre-differentiated for 5 days to obtain a chondrocyte phenotype. See figure below (in the Response to Reviewers attached file), these data are now added to the manuscript as Supplemental Figure 2. In addition to show more confidence in our NKX3-2 siRNA data generated in the SW1353 cells, we repeated the majority of the experiments in human primary chondrocytes (3 independent donors) using the same human NKX3-2 siRNA duplex and obtained similar results. These data are now added to the manuscript (Figure 3G and 4C). 

To meet the Reviewer’s second request for specificity control, we performed the requested experiment using the NKX3-2 siRNA and re-expression of NKX3-2 using a FLAG-NKX3-2 expression vector with optimized codon-usage that evades the NKX3-2 siRNA-mediated silencing. Overexpression of NKX3-2 by transfecting SW1353 cells or human primary chondrocytes (n=3 independent donors) with the FLAG‐NKX3-2 vector [1] resulted in a significantly decreased expression of RUNX2 accompanied by an increased expression of rRNAs and accompanying translational capacity in these cells. These data are reciprocal to the NKX3-2 knockdown (also presented in Figure 3) and in line with the BMP7-mediated increase in rRNA expression and translation capacity from Figure 1. These effects could not be significantly altered by the NKX3-2 siRNA (see Figure below in the Response to Reviewers attached file). We hope that these data provide ample additional insight for the Reviewer on the role of NKX3-2 in chondrocyte rRNA expression and translational capacity. The NKX3-2 overexpression data is now included in the main manuscript as Figure 4 and the whole figure as presented below in the Supplemental files accompanying this manuscript. 

 

S2 Fig: BMP7-induced rRNA levels are NKX3-2 dependent in ATDC5 cells 

ATDC5 cells (6.400 cells/cm2) were differentiated in the chondrogenic lineage for 6 days to acquire a chondrocyte phenotype and then transfected (according to the manufacturer’s protocol, using HiPerfect, Qiagen) with either a scrambled (Control RNAi (Eurogentec)) or Nkx3-2 (Nkx3-2 RNAi) siRNA duplex (100nM; sense: 5’-CAGAGACGCAAGUGAAGAUTT-3’, anti-sense: 5’-AUCUUCACUUGCGUCUCUGTT-3’) and exposed to BMP7 (1nM) for 24 hours after which expression levels of Nkx3-2, Runx2, 18S rRNA, 5.8S rRNA, 28S rRNA, Ubtf and Tcof1 were measured by RT-qPCR. Data were normalized to cyclophilin expression and set relative to the control condition (n=3 samples per condition). Differentiation medium for ATDC5 consisted of Dulbecco's minimal essential medium (DMEM)/F12 (Invitrogen), 5% fetal calf serum (FCS) (Sigma-Aldrich), 1% antibiotic/antimycotic (Invitrogen) and 1% NEAA (non-essential amino acids; Invitrogen), 10mg/ml insulin (Sigma-Aldrich),10mg/ml transferrin (Roche) and 30 nM sodium selenite (Sigma-Aldrich). Statistical significance was determined using a two-tailed unpaired Student’s t-tests. Bars show the mean ±SEM. * P<0.05, ** P<0.01, *** P<0.001. ns= not significant.

 

S3 Fig: NKX3-2 overexpression increases rRNA levels and is associated with increased translational capacity

A-B. NKX3-2 was overexpressed by transient transfection of a codon usage optimized FLAG-NKX3-2 vector into SW1353 cells and FLAG-empty vector was used as a negative control. The next day, these cells were transfected with either a scrambled (Control RNAi) or NKX3-2 (NKX3-2 RNAi) siRNA duplex (100nM) and after 24 hours cells were harvested. Expression of mRNA for the indicated genes was determined by real-time RT-qPCR. Data were normalized to cyclophilin expression and set relative to the control condition (A-B: n=3 samples per condition). C. Translational capacity was determined and Puromycilation data were normalized to DNA content and calculated relative to the control condition (n=5 samples per condition). Statistical significance was determined using unpaired two-tailed Student’s t-tests. Bars show the mean ±SEM. * P<0.05, ** P<0.01, *** P<0.001 versus control conditions. ns = not significant. 

References 

1. Caron MM, Emans PJ, Surtel DA, van der Kraan PM, van Rhijn LW, Welting TJ. BAPX-1/NKX-3.2 acts as a chondrocyte hypertrophy molecular switch in osteoarthritis. Arthritis Rheumatol. 2015;67(11):2944-56. Epub 2015/08/08. doi: 10.1002/art.39293. PubMed PMID: 26245691.

---

## [Decision Letter · Decision Letter 2]

20 Jan 2022

BMP7 increases protein synthesis in SW1353 cells and determines rRNA levels in a NKX3-2-dependent manner

PONE-D-20-37112R2

Dear Dr. Caron,

We’re pleased to inform you that your manuscript has been judged scientifically suitable for publication and will be formally accepted for publication once it meets all outstanding technical requirements.

Kind regards,

Andre van Wijnen

Academic Editor

PLOS ONE

Additional Editor Comments (optional):

Reviewers' comments:

Reviewer's Responses to Questions

**Comments to the Author**

1. If the authors have adequately addressed your comments raised in a previous round of review and you feel that this manuscript is now acceptable for publication, you may indicate that here to bypass the “Comments to the Author” section, enter your conflict of interest statement in the “Confidential to Editor” section, and submit your "Accept" recommendation.

Reviewer #1: All comments have been addressed

2. Is the manuscript technically sound, and do the data support the conclusions?

Reviewer #1: Yes

3. Has the statistical analysis been performed appropriately and rigorously? 

Reviewer #1: Yes

4. Have the authors made all data underlying the findings in their manuscript fully available?

Reviewer #1: Yes

5. Is the manuscript presented in an intelligible fashion and written in standard English?

Reviewer #1: Yes

6. Review Comments to the Author

Reviewer #1: The revision has addressed my earlier concerns. The manuscript meets the requirements for acceptance to PLoS One.

7. PLOS authors have the option to publish the peer review history of their article (what does this mean?). If published, this will include your full peer review and any attached files.

Reviewer #1: No

---

## [Editor Report · Acceptance letter]

31 Jan 2022

PONE-D-20-37112R2 

BMP7 increases protein synthesis in SW1353 cells and determines rRNA levels in a NKX3-2-dependent manner 

Dear Dr. Caron:

I'm pleased to inform you that your manuscript has been deemed suitable for publication in PLOS ONE. Congratulations! Your manuscript is now with our production department. 

Kind regards, 

on behalf of

Dr. Andre van Wijnen 

Academic Editor

PLOS ONE